# OpenReview forum: "Did You Faithfully Say What You Thought? Bridging the Gap Between LLMs Neural Activity and Self-Explanations"
_ICLR.cc/2026/Conference — Submitted to ICLR 2026_

### Official Review · Reviewer_2wXs · 2025-10-18

**Soundness:** 2
**Presentation:** 3
**Contribution:** 3
**Rating:** 4
**Confidence:** 4

**Summary:**

The paper introduces NeuroFaith, a way to measure faithfulness of LLM self-explanations (in natural language). NeuroFaith identifies concepts mentioned in a model’s explanation and tests whether these concepts actually influence the model’s predictions through mechanistic analysis of internal activations. The framework is instantiated for both 2-hop reasoning and classification tasks, using techniques like Patchscopes to quantify alignment between explanations and internal reasoning. Experiments on Gemma-2 models show mixed results, but generally that larger or more accurate models tend to produce more faithful explanations. Finally, the authors use NeuroFaith to label examples for high and low faithfulness, compute steering vectors from that to increase LLM faithfulness.

**Strengths:**

* The paper is well-motivated, clearly identifying the limitations of current faithfulness evaluation methods and convincingly taking related work's argument that assessing faithfulness requires examining the model’s internal neural representations, not just behavioral agreement. This makes NeuroFaith is conceptually elegant, combining interpretability and faithfulness measurement in a unified, mechanistic approach that grounds explanations in internal model activity.
* The experiments cover multiple model scales (Gemma-2 2B–27B), which provides evidence into how faithfulness increases with scale.
* The faithfulness steering sections go beyond evaluation to show practical ways to improve explanation faithfulness at inference time.

**Weaknesses:**

* **Lack of metric validation:** The paper does not empirically validate how well NeuroFaith correlates with existing faithfulness measures or human judgments of explanation quality. It would strengthen the claim of “measuring faithfulness” to compare against recent dedicated ways to even automatically test how much we can trust faithfulness metrics: https://arxiv.org/abs/2502.18848
* **Not comparing against other faithfulness metrics:** The evaluation relies primarily on custom experimental setups rather than established benchmarks (e.g., Parcalabescu & Frank, 2024 cited by this paper or https://arxiv.org/abs/2502.18848), making it difficult to assess how NeuroFaith performs relative to prior metrics. This continues the trend of isolated metrics without direct comparison, which limits the interpretability of reported results.
* **Weak model type coverage:** The study is restricted to the Gemma-2 model family (2B–27B), leaving open how general the findings are to other architectures or training paradigms (e.g., LLaMA, Mistral, Vicuna, Qwen3, Deepseek, etc).
* **Unstudied effect of the interpretability method**: The framework’s results likely depend on the chosen interpretability method, but the paper does not systematically analyze this dependency or the potential unfaithfulness of these underlying tools.
* **Unquantified effect of llm as a judge:** The reliance on LLM-as-a-Judge for concept extraction could introduce biases that propagate into the faithfulness score, yet the paper only briefly acknowledges this limitation without quantifying its impact.

**Questions:**

Do you believe that activation steering alone is sufficient for improving faithfulness, or would it be worthwhile to explore reinforcement learning or fine-tuning using NeuroFaith as a reward signal?


Suggestion: In the abstract you use passive voice in "Additionally, a linear faithfulness probe based on NeuroFaith is developed to detect unfaithful self-explanations from representation space and improve faithfulness through steering." It makes it unclear whether you did this, or someone else or whether this is left to future work. Passive voice is bad writing style because it hides the actant.

---

> ### Author Response · Authors · 2025-11-25
> **Answer to reviewer 2wXs**
>
> We thank the reviewer for their time, thoughtful consideration, and constructive feedback.
>
> ## Weaknesses
>
> > Lack of metric validation and Not comparing against other faithfulness metrics
>
> We thank the reviewer for these important comments. We actually conducted several experiments to compare our framework against the widely used Counterfactual Intervention (CI) methodology [1] for 2-hop reasoning in Appendix F (page 34). In this section, we propose a protocol to compare faithfulness measures based on their ability to: (1) help understand LLM errors and debug them, and (2) detect answers based on incorrect reasoning chains. Our approach systematically outperforms CI across all models (see Table 12, page 35), highlighting that CI can be overly permissive in 2-hop reasoning scenarios.
>
> > Weak model type coverage
> We thank the reviewer for this relevant comment. We have now applied our framework to **Mistral-7B-Instruct-v0.3**, obtaining the following results:
>
> | Model | Judge | Self-NLE correctness |  | Latent Hop 1 correctness |  | Self-NLE faithfulness |  |
> |---|---|---|---|---|---|---|---|
> |  |  | accurate | inaccurate | accurate | inaccurate | accurate | inaccurate |
> | mistral-0.3-7B | qwen | 70.20% | 37.90% | 52.90% | 29.60% | 51.10% | 32.60% |
>
> Comparing these results to gemma-2-9B (similar size) using the detailed taxonomy introduced in Appendix E, we observe interesting architectural differences:
>
> | Model | Judge | C1 | C2 | C3 | C4 | C5 | C6 | C7 | C8 | C9 | C10 |
> |---|---|---|---|---|---|---|---|---|---|---|---|
> | mistral-0.3-7B | qwen | 46.50% | 13.70% | 7.10% | 8.40% | 24.20% | 20.60% | 4.00% | 19.30% | 5.20% | 50.90% |
> | gemma-2-9B | qwen | 26.10% | 4.50% | 14.50% | 14.10% | 40.90% | 14.90% | 4.20% | 20.00% | 6.80% | 54.00% |
>
> Mistral-7B-Instruct-v0.3 and Gemma-2-9B show similar propensity to generate faithful self-explanations for accurate predictions (C6-C10: ~50-54%), but Mistral generates significantly less faithful explanations for inaccurate predictions (C1-C5). This suggests architecture-specific failure modes that NeuroFaith can successfully identify.
>
> > Unquantified effect of llm as a judge
>
> We thank reviewer for this relevant comment. We agree that our framework depends on an auxiliary model to extract concepts from self-explanations. To address concerns about potential biases, we conducted additional experiments using a different LLM annotator (Phi-4) on the 2-hop reasoning task:
>
> | Model | Judge | Self-NLE correctness |  | Latent Hop 1 correctness |  | Self-NLE faithfulness |  | Faithfulness Corr w/ initial |
> |---|---|---|---|---|---|---|---|---|
> |  |  | accurate | inaccurate | accurate | inaccurate | accurate | inaccurate |  |
> | gemma-2-2B | Phi-4 | 57.80% | 56.50% | 47.50% | 48.30% | 48.00% | 56.50% | 86.50% |
> | gemma-2-9B | Phi-4 | 73.70% | 55.10% | 58.50% | 44.30% | 61.70% | 54.80% | 91.20% |
> | gemma-2-27E | Phi-4 | 77.80% | 55.70% | 64.90% | 46.90% | 69.10% | 59.80% | 90.90% |
>
> **These results align closely with those obtained in the paper** using Qwen-32B as the auxiliary LLM (see Table 1 in the paper). The **Faithfulness Corr w/ initial column** above represents the correlation between faithfulness scores obtained with Qwen-32B and Phi-4, ranging between 86% and 88%.
>
> We also conducted additional experiments to validate Qwen-3-32B's reliability as a judge. For 2-hop reasoning, we used the same prompt as in the paper to retrieve bridge objects on complete reasoning chains (o₁,r₁,o₂), (o₂,r₂,o₃), **Qwen-3-32B successfully detected the expected bridge object 99.5% of the time**. For classification, we randomly selected 3 concepts to assess detection reliability based on ground truth from [1]. Qwen-3-32B detected concepts with 80% and 85% accuracy for Ledgar and AGNews, respectively.
>
> These results demonstrate that: (1) our framework is robust to the choice of judge for concept retrieval, and (2) the employed judge is reliable for concept detection.
>
> ## Questions
>
> > Do you believe that activation steering alone is sufficient for improving faithfulness, or would it be worthwhile to explore reinforcement learning [...]
>
> Thank you for this excellent question. We believe activation steering demonstrates that faithfulness is a manipulable property encoded in representation space, but we strongly agree that reinforcement learning (GRPO) approaches would be valuable for more robust improvements.
>
> > Suggesion
>
> We thank the reviewer for this excellent stylistic suggestion. We will revise the abstract to clearly highlight our contribution.
>
> ## References
>
> [1] Atanasova, P., Camburu, O. M., Lioma, C., Lukasiewicz, T., Simonsen, J. G., & Augenstein, I. (2023, July). Faithfulness Tests for Natural Language Explanations. In Proceedings of the 61st Annual Meeting of the Association for Computational Linguistics (Volume 2: Short Papers) (pp. 283-294).

---

> > ### Comment · Reviewer_2wXs · 2025-11-26
> > **Response**
> >
> > Thank you for all these additions! I am increasing my score from 4 to 6.
> >
> > I cannot further increase my score at this point because, although Appendix F already provides comparisons against other metrics, it does not evaluate whether the metric actually tests for faithfulness in the way that, for example, the evaluation in https://arxiv.org/abs/2502.18848 does. Such a test is essential: even if two faithfulness metrics correlate, they may still fail to measure the underlying construct they are intended to capture.

---

### Official Review · Reviewer_DA1R · 2025-10-30

**Soundness:** 2
**Presentation:** 4
**Contribution:** 2
**Rating:** 4
**Confidence:** 4

**Summary:**

A framework is proposed where self-generated natural language explanations of models are evaluated on their faithfulness through interpretability techniques. Specifically it is proposed to extract from each explanation a set of concepts that it evokes; then, one of a variety of interpretability techniques can be used to see if those concepts are present in the latent states of the model. The degree to which they are present is used to estimate faithfulness.

A number of experiments are performed to show how this framework can be instantiated. These experiments generally show that: (1) self-explanations are more faithful when they come from larger models, (2) explanations of accurate predictions (as opposed to inaccurate ones) are more faithful, and (3) faithfulness depends on task.

This measure of faithfulness is further used to look for a direction in the models' latent spaces which correlates with it. Such directions are found, and depending on the task, they predict faithfulness with F1 scores roughly between 60 and 75.
Finally, the directions are used to steer the model towards producing more faithful explanations.

**Strengths:**

The use of interpretability techniques to evaluate the faithfulness of (self-)explanations makes a lot of sense, and the paper makes sensible choices in pursuing this.

The presentation is generally good, clear language, diagrams, tables, figures, and sensible sectioning.

High information density, multiple experiments using different interpretability techniques to showcase the versatility of the framework.

**Weaknesses:**

Lines 457-459 state that 'Direct faithfulness amplification consistently outperforms hallucination and deceptivenes inhibition', but from Table 3 that looks like the wrong take-away: Hallucination inhibition performs very similarly to faithfulness amplification, especially for the smaller models. Looking at the cosine similarities in the appendix, we see something similar: for smaller models the faithfulness direction is not negatively correlated to the same extent. That also seems worth mentioning in the paragraph of lines 409-431.

Potentially some information missing (see question 2).

**Questions:**

1. Does the statistical significance calculation for Figure 3 take into account the fact that you are selecting (taking max) over many different layers, thereby increasing the chances of getting lucky?

2. Is the steering done in the layer where the faithfulness direction was most predictive of faithfulness? or some other selection of layers? What about tokens, is the steering vector added to all tokens? What about the hallucination and deceptiveness baselines?

3. What data is used for calculating the concept-activation vectors described in lines 314-323? Generally speaking, do we always have enough of this data for each concept that might be detected? Also, a fixed set of concepts $\mathcal{C}$ is assumed, where do these come from?

---

> ### Author Response · Authors · 2025-11-24
> **Answer to reviewer DA1R [1]**
>
> We thank the reviewer for their time, thoughtful consideration, and constructive feedback.
>
> ## Weaknesses
>
> > [...] that looks like the wrong take-away: Hallucination inhibition performs very similarly to faithfulness amplification [...]
>
> We appreciate the reviewer raising this important point. We agree that our original framing may have overemphasized distinctions between these approaches. We will revise the manuscript to more accurately reflect that hallucination inhibition and faithfulness amplification achieve comparable performance, and clarify the appropriate interpretation of these results.
>
> ## Questions
>
> > Does the statistical significance calculation for Figure 3 take into account the fact that you are selecting (taking max) over many different layers [...]
>
> Thank you for this insightful observation. You are correct that our initial statistical analysis did not properly account for multiple comparisons when applying the "max" operator across layers. We have re-run our statistical analysis with Bonferroni correction to address this issue. The corrected results now align closely with the statistical significance patterns observed using the "mean" operator. We will update Figure 3 and the corresponding text to reflect these corrected statistical tests.
>
> > Is the steering done in the layer where the faithfulness direction was most predictive of faithfulness? [...]
>
> Thank you for seeking clarification on our intervention methodology. Yes, steering was applied specifically at layers where faithfulness is reliably encoded, as stated in lines 452-453: *"Faithfulness intervention targets only layers with F1 > 60%, ensuring modifications occur where faithfulness is reliably encoded."* The intervention was applied to the last token of the sequence. We followed parallel principles for hallucination and deceptiveness mitigation. For identifying intervention layers for deceptiveness mitigation specifically, we relied on the layer-wise performance results reported in [1], which demonstrated optimal probe performance at intermediate layers (layers 20-25 for Llama models).
>
> > What data is used for calculating the concept-activation vectors described in lines 314-323 [...]
>
> Thank you for requesting clarification on our CAV computation methodology. We utilized the concept sets and annotated datasets directly from [2], which provide both concept definitions and labeled examples. Specifically, as stated in line 360: *"We use the concept set C and labels of AGNews and Ledgar from Bhan et al. (2025) to compute the CAVs."* This ensures our concept activation vectors are computed using established, validated concept representations from prior work.
>
> ## References
>
> [1] Rimsky, N., Gabrieli, N., Schulz, J., Tong, M., Hubinger, E., & Turner, A. (2024, August). Steering llama 2 via contrastive activation addition. In Proceedings of the 62nd Annual Meeting of the Association for Computational Linguistics (Volume 1: Long Papers) (pp. 15504-15522).
>
> [2] Milan Bhan, Yann Choho, Jean-Noël Vittaut, Nicolas Chesneau, Pierre Moreau, and Marie-Jeanne Lesot. 2025. Towards Achieving Concept Completeness for Textual Concept Bottleneck Models. In Findings of the Association for Computational Linguistics: EMNLP 2025, pages 2007–2024, Suzhou, China. Association for Computational Linguistics.

---

> > ### Comment · Reviewer_DA1R · 2025-11-27
> >
> > Thank you for your response, I have raised my score, trusting you will appropriately incorporate your clarifications in the paper.

---

### Official Review · Reviewer_LtSp · 2025-10-31

**Soundness:** 2
**Presentation:** 1
**Contribution:** 3
**Rating:** 2
**Confidence:** 3

**Summary:**

The paper studies faithfulness: do the natural language explanations (NLEs) models produce for their own decisions actually reflect their true reasoning process?

The paper uses methods based on internal model representations to determine whether certain concepts are represented (via probing) and/or important (via representation engineering). In particular, the paper runs two sets of experiments:

2-hop reasoning: the paper evaluates three Gemma models on a 2-hop reasoning task. It assesses whether models contain latent representations of these bridge entities while solving the questions, and also whether self-NLEs contain these bridge entities.

Classification: the paper evaluates three Gemma models on a classification task, and has them produce self-NLEs. It extracts relevant concepts from these NLEs using Qwen3-32B. It then determines the causal influence of the concepts by erasing their vectors during forward propagation, to determine whether the concept was actually necessary to complete the task.

Finally, the paper tests interventions for improving model faithfulness: steering models toward faithfulness vectors determined by the paper’s own faithfulness evaluations, as well as vectors for hallucination and deceptiveness from prior work.

**Strengths:**

1) Using mechanistic methods to determine faithfulness is a promising approach: it has the potential to give a better view into the model’s actual decisionmaking process, so that we can check whether explanations are accurately representing this process.

2) The paper combines a number of interesting techniques from the mechanistic interpretability literature in novel and creative ways.

3) The steering results are interesting: it would be very cool if there was a linear vector which could generally increase model faithfulness! Evaluating the faithfulness vector on the same tasks used to determine its direction risks Goodharting, but it’s a cool result that steering away from hallucination and deceptiveness vectors from prior work also improves performance on the paper’s own faithfulness metrics.

**Weaknesses:**

1) I often found it hard to figure out exactly what the paper’s experiments are doing. For example, Section 3 is highly abstract, and I came away not being sure what had actually been described, other than “extract concepts and measure their presence and/or importance mechanistically”. I think the formality and notation in this section is obfuscating more than it’s clarifying.

This also makes section 4 and 5 hard to follow, since they need to spend time explaining how their components map to the notation described in section 3, instead of just explaining how the specific instantiation works. The paper might be clearer by presenting the two methods more independently, rather than attempting to define the NeuroFaith framework to encapsulate both of them.

2) The assumption behind the 2-hop experiments is that “For f to correctly answer, it must internally compute the bridge object during the first hop before executing the second hop.” This isn’t always true: LLMs may develop shortcuts, e.g. by encountering head and answer entities in the same sequence, or guessing the answer with frequency-based priors. See Yang et al. 2025, e.g. Table 2 showing that some datasets can have very high shortcut prevalence (https://arxiv.org/abs/2411.16679).

But if a model ends up with the correct answer via a shortcut, rather than via the bridge entity, self-NLE “faithfulness” isn’t really a faithfulness metric, since the true faithful explanation of the model’s internal process would be the shortcut.

Indeed, Table 1 from the authors’ paper shows that for Gemma models, when the model’s prediction is accurate, “Latent Hop 1 Correctness” is between 48%-65%. IIUC, this means that in 35-52% of examples where the model ends up with the correct answer, the mechanistic interpretability method is unable to identify coordinates in the circuit such that the natural language description of those coordinates include the concept of interest. This seems consistent with frequent shortcut usage. (Though it’s also consistent with a failure of the latent extraction method to correctly identify feature usage.)

If you want to rely on models actually depending on bridge entities, you should evaluate using a dataset which explicitly filters out shortcuts, e.g. SOCRATES (https://github.com/google-deepmind/latent-multi-hop-reasoning). This would also give more signal on whether the latent extraction method is functioning correctly.

3) The 2-hop results use probing to detect concept presence. But probing alone doesn’t show causation: a bridge entity might be present in the residual stream, but ignored by later layers in favor of a shortcut. If the goal is to determine the model’s actual process, why use probing rather than concept importance?

4) The paper’s experiments on classification rely on Qwen3-32B to extract concepts from the self-NLE, but does it actually perform this job reliably?

More generally, the paper’s methods make use of a number of components, but it’s unclear to what extent we can trust each component. If the paper’s methods give surprising results, can we trust this to be a genuine finding? Or could it just be a failure of one of the individual components, e.g. the LLM doing concept extraction? How could we tell?

5) The paper’s tables and figures are mostly lacking confidence intervals or statistical tests (other than Figure 3), so it’s unclear which group differences are significant and which may be noise.

**Questions:**

1) “We leverage this information to define circuit Γ and generate the natural language description  ̃hℓk of hidden state hℓk” - how are you generating this natural language description? I’ve reread paragraph “Probing-based Concept Interpretation” but I’m having difficulty figuring out which specific method you’re using.

2) The classification experiments extract concepts from the self-NLE via Qwen3, but how were these mapped to internal representations? If you’re using a concept mapping from existing work, can you explain the form of that mapping, and how it was generated?

More generally, there are a lot of components being used from prior work; I think it would help readability to give more description of these methods.

3) The paper provides examples in appendices G and H; I think it would substantially clarify the presentation to include at least one example for each methodology in the main paper.

4) I’m struggling to understand the “Importance-based Concept Interpretation” paragraph: how are you actually arriving at the score determining whether a given example is faithful or not?

---

> ### Author Response · Authors · 2025-11-26
> **Answer to reviewer LtSp (1/2)**
>
> We thank the reviewer for their time, thoughtful consideration, and constructive reviews.
>
> ## Weaknesses
>
> > [...] LLMs may develop shortcuts, e.g. by encountering head and answer entities in the same sequence [...]
>
> Thank you for this highly relevant comment. We fully agree that LLMs can address 2-hop reasoning through shortcuts. Our detailed taxonomy explicitly integrates this possibility (see Appendix E, page 28, Category C6: "Shortcut Learning"). We estimate that Gemma-2 models perform shortcut learning in 12.8-27.8% of correct predictions, depending on model size (Table 8, page 29).
>
> This phenomenon is also discussed in the main body (lines 298-304): "*[...] This suggests that models can arrive at correct answers through alternative reasoning pathways that bypass explicit bridge object computation. The models may be leveraging (1) direct associations between source and target entities (shortcut learning), or (2) alternative pathways making the model accurately answer for a reason different than the ground truth bridge object (see category 9 in Figure 25).*"We will make sure to clarify this point in the next version of the paper.
>
> > [...] why use probing rather than concept importance? [...]
>
> Thank you for highlighting this crucial methodological choice. We agree that concept importance methods provide stronger causal guarantees than probing.
>
> However, we selected Patchscopes [1] (a non-linear probe) for 2-hop reasoning for the following reasons: (1) Linear encoding assumption: Concept importance methods based on gradients (TCAV [2]) or representation engineering [3] require concepts to be linearly encoded in representation space. Bridge objects in 2-hop reasoning are not necessarily linearly separable, limiting the applicability of these methods. (2) We followed existing work on sequential latent reasoning in multi-hop tasks [4], which successfully employs non-linear probing methods like Patchscopes to decode intermediate reasoning steps.
>
> Importantly, our framework is interpreter-agnostic. For tasks where concepts are linearly encoded (verified via F1 > 60% with linear probes), we do employ concept importance methods (e.g. Section 5), demonstrating that our approach adapts the interpretation method to task characteristics.
>
> > [...] rely on Qwen3-32B to extract concepts from the self-NLE, but does it actually perform this job reliably?
>
> Thank you for highlighting this crucial point. We conducted additional experiments to validate Qwen-3-32B's reliability as a judge. For 2-hop reasoning, we used the same prompt as in the paper to retrieve bridge objects on complete reasoning chains (o₁,r₁,o₂), (o₂,r₂,o₃), **Qwen-3-32B successfully detected the expected bridge object 99.5% of the time**. For classification, we randomly selected 3 concepts to assess concept detection accuracy based on ground truth from [5]. Qwen-3-32B detected concepts with 80% and 85% accuracy for Ledgar and AGNews, respectively.
>
> We also tested Phi-4 as an alternative judge, finding 86-91% correlation with Qwen-32B's faithfulness scores, demonstrating that our results are robust to judge selection. We believe that these additional experiments demonstrate that Qwen-3-32B is a reliable concept extractor for our framework.
>
> > [...] If the paper’s methods give surprising results, can we trust this to be a genuine finding?
>
> Thank you for this important question. We fully agree that faithfulness measurement faces the fundamental challenge of lacking ground truth for the model's "true" reasoning process.
>
> To address this, we developed a validation protocol in Appendix F (pages 33-36) that evaluates faithfulness measures based on their practical utility for: (1) Helping understand and debug LLM failures and (2) Detecting correct answers based on incorrect reasoning chains. We show that our approach systematically outperforms the widely used Counterfactual Intervention (CI) [6] faithfulness measurement method across all models (see Table 12, page 35) and gives strong results overal (see Table 13, page 35).
>
> We believe this experimetal results strenghten the validity of our faithfulness measurement approach.

---

> ### Author Response · Authors · 2025-11-26
> **Answer to reviewer LtSp (2/2)**
>
> ## Questions
>
> > [...] how are you generating this natural language description?
>
> Than you for this question. We used Patchscope to generate natural language interpretation of LLMs' hidden states. This is stated line 245-246: "*We employ natural language interpretation methods such as Selfie and Patchscopes, rather than linear probes, as they provide higher accuracy and are unsupervised [...]*". Additional implementation details are given in Appendix, Section D.2, page 18.
>
> > [...] but how were these (concepts) mapped to internal representations? [...]
>
> Thank you for this clarification request. We want to clarify that we do not perform concept-to-representation mapping for 2-hop reasoning. We used Qwen-3 as annotator by using the prompts given in Appendix D.1, pages 16-17.
>
> > [...] how are you actually arriving at the score determining whether a given example is faithful or not?
>
> Thank you for this question. Importance-based Concept Interpretation evaluates whether the concepts mentioned in a self-explanation actually influence the model's prediction. The methodology works as follows:(1) We identify all concepts mentioned in the self-explanation, (2) For each concept, we test its importance by erasing it from the model's internal representations during inference and measuring how much the prediction changes and (3) We compute the faithfulness score as the percentage of mentioned concepts that prove to be important (i.e., that significantly affect the prediction when erased). This approach uses established techniques from representation engineering [3] and concept erasure [7] to quantify each concept's causal influence on the model's decision-making process.
>
> ## References
>
> [1] Ghandeharioun, A., Caciularu, A., Pearce, A., Dixon, L., & Geva, M. (2024, July). Patchscopes: A Unifying Framework for Inspecting Hidden Representations of Language Models. In International Conference on Machine Learning (pp. 15466-15490). PMLR.
>
> [2] Kim, B., Wattenberg, M., Gilmer, J., Cai, C., Wexler, J., & Viegas, F. (2018, July). Interpretability beyond feature attribution: Quantitative testing with concept activation vectors (tcav). In International conference on machine learning (pp. 2668-2677). PMLR.
>
> [3] Zou, A., Phan, L., Chen, S., Campbell, J., Guo, P., Ren, R., ... & Hendrycks, D. (2023). Representation engineering: A top-down approach to ai transparency. arXiv preprint arXiv:2310.01405.
>
> [4] Biran, E., Gottesman, D., Yang, S., Geva, M., & Globerson, A. (2024, November). Hopping Too Late: Exploring the Limitations of Large Language Models on Multi-Hop Queries. In Proceedings of the 2024 Conference on Empirical Methods in Natural Language Processing (pp. 14113-14130).
>
> [5] Milan Bhan, Yann Choho, Jean-Noël Vittaut, Nicolas Chesneau, Pierre Moreau, and Marie-Jeanne Lesot. 2025. Towards Achieving Concept Completeness for Textual Concept Bottleneck Models. In Findings of the Association for Computational Linguistics: EMNLP 2025, pages 2007–2024, Suzhou, China. Association for Computational Linguistics.
>
> [6] Atanasova, P., Camburu, O. M., Lioma, C., Lukasiewicz, T., Simonsen, J. G., & Augenstein, I. (2023, July). Faithfulness Tests for Natural Language Explanations. In Proceedings of the 61st Annual Meeting of the Association for Computational Linguistics (Volume 2: Short Papers) (pp. 283-294).
>
> [7] Belrose, N., Schneider-Joseph, D., Ravfogel, S., Cotterell, R., Raff, E., & Biderman, S. (2023). Leace: Perfect linear concept erasure in closed form. Advances in Neural Information Processing Systems, 36, 66044-66063.

---

### Official Review · Reviewer_sWcB · 2025-11-08

**Soundness:** 2
**Presentation:** 2
**Contribution:** 2
**Rating:** 4
**Confidence:** 3

**Summary:**

This paper proposes NeuroFaith for evaluating the faithfulness of self-NLEs from large language models. The authors demonstrate this framework's flexibiity on both 2-hop reasoning and classification tasks. They then successfully manipulate using activation steering to improve explanation faithfulness at inference time, offering a path toward more transparent models.

**Strengths:**

1.The authors propose a reasonable faithfulness definition, checking if the model's explanation aligns with its actual computation and successfully improve the model's faithfulness via activation steering.

2.The framework is successfully applied to both 2 hop reasoning tasks and classification tasks.

**Weaknesses:**

As mentioned in conclusion, the faithfulness score depends on what the auxiliary model considers a concept, which could introduce biases or blind spots. The authors also concede that circuit selection 'requires domain expertise or expensive circuit discovery', which undermines the framework's scalability and ease of application to new tasks

**Questions:**

1.Did the authors measure the corresponding change in task accuracy for the steered examples? Does this intervention also happen to correct the final prediction, or does it merely make the explanation for the wrong answer more faithful?

2.How sensitive are your final faithfulness scores to the choice of the auxiliary LLM used for concept extraction? If you were to use a different judge, how much would the F(x, e) scores and the resulting steering vector change?

---

> ### Author Response · Authors · 2025-11-25
> **Rebuttal**
>
> We thank the reviewer for their time, thoughtful consideration, and constructive feedback.
>
> ## Weaknesses
>
> > 1. [...] the auxiliary model considers a concept, which could introduce biases
>
> We thank the reviewer for this relevant comment. We agree that our framework depends on an auxiliary model to extract concepts from self-explanations. To address concerns about potential biases, we conducted additional experiments using a different LLM annotator (Phi-4) on the 2-hop reasoning task:
>
> | Model | Judge | Self-NLE correctness |  | Latent Hop 1 correctness |  | Self-NLE faithfulness |  | Faithfulness Corr w/ initial |
> |---|---|---|---|---|---|---|---|---|
> |  |  | accurate | inaccurate | accurate | inaccurate | accurate | inaccurate |  |
> | gemma-2-2B | Phi-4 | 57.80% | 56.50% | 47.50% | 48.30% | 48.00% | 56.50% | 86.50% |
> | gemma-2-9B | Phi-4 | 73.70% | 55.10% | 58.50% | 44.30% | 61.70% | 54.80% | 91.20% |
> | gemma-2-27E | Phi-4 | 77.80% | 55.70% | 64.90% | 46.90% | 69.10% | 59.80% | 90.90% |
>
> **These results align closely with those obtained in the paper** using Qwen-32B as the auxiliary LLM (see Table 1 in the paper). The **Faithfulness Corr w/ initial column** above represents the correlation between faithfulness scores obtained with Qwen-32B and Phi-4, ranging between 86% and 88%.
>
> We also conducted additional experiments to validate Qwen-3-32B's reliability as a judge. For 2-hop reasoning, we using the same prompt as in the paper to retrieve bridge objects on complete reasoning chains (o₁,r₁,o₂), (o₂,r₂,o₃), **Qwen-3-32B successfully detected the expected bridge object 99.5% of the time**. For classification, we randomly selected 3 concepts to assess detection reliability based on ground truth from [1]. Qwen-3-32B detected concepts with 80% and 85% accuracy for Ledgar and AGNews, respectively.
>
> These results demonstrate that: (1) our framework is robust to the choice of judge for concept retrieval, and (2) the employed judge is reliable for concept detection.
>
> > 2. [...] circuit selection 'requires domain expertise or expensive circuit discovery', which undermines the framework's scalability [...]
>
> Thank you for this comment. We acknowledge that circuit discovery can be viewed as a potential bottleneck for certain tasks. However, a practical and scalable alternative is to focus on the residual stream at the last token of the sequence, as adopted by many other works such as [2] and [3]. This approach significantly reduces the need for task-specific circuit discovery while maintaining effectiveness. In our experiments, this simpler approach provides meaningful faithfulness measurements without requiring extensive domain expertise or computational resources for circuit discovery.
>
> ## Questions
>
> > 1. Did the authors measure the corresponding change in task accuracy for the steered examples? [...]
>
> We thank the reviewer for this important question. Since we perform activation steering after the answer has been generated (following a predict-then-explain protocol), there is no direct impact on task accuracy. The steering intervention specifically targets the faithfulness of the generated explanation rather than the initial prediction.
>
> >2.  How sensitive are your final faithfulness scores to the choice of the auxiliary LLM used for concept extraction? [...]
>
> Please see our response to Weakness 1
>
> ## References
>
> [1] Milan Bhan, Yann Choho, Jean-Noël Vittaut, Nicolas Chesneau, Pierre Moreau, and Marie-Jeanne Lesot. 2025. Towards Achieving Concept Completeness for Textual Concept Bottleneck Models. In Findings of the Association for Computational Linguistics: EMNLP 2025, pages 2007–2024, Suzhou, China. Association for Computational Linguistics.
>
> [2] Li, K., Patel, O., Viégas, F., Pfister, H., & Wattenberg, M. (2023). Inference-time intervention: Eliciting truthful answers from a language model. Advances in Neural Information Processing Systems, 36, 41451-41530.
>
> [3] Ghandeharioun, A., Caciularu, A., Pearce, A., Dixon, L., & Geva, M. (2024, July). Patchscopes: A Unifying Framework for Inspecting Hidden Representations of Language Models. In International Conference on Machine Learning (pp. 15466-15490). PMLR.

---

### Meta-Review · Area_Chair_38qx · 2026-01-10

**Summary:**

This paper introduces NeuroFaith, a mechanistic framework for evaluating the faithfulness of LLM self-explanations by extracting concepts from explanations and testing whether those concepts causally influence the model's prediction via interpretability tools. The approach is demonstrated on 2-hop reasoning and classification, and is further used to learn linear faithfulness probes and to improve explanation faithfulness via steering.

The reviewers generally appreciate the importance of the problem tackled by this paper, as well as the fact that the approach pushes beyond purely behavioral proxies by considering internal representations and interventions. At the same time, the reviewers have raised concerns mostly around the validity of the actual construction: bias of auxiliary LLM, stronger causality guarantees, correlation with human judgement, comparison with other metrics.

**Reviewer Concerns:**

The experiments are demonstrated on a broad set of tasks and include steering. The rebuttal provides meaningful additions like using an additional LLM annotator and extending model coverage, and this has resulted in some scores to be raised upwards.

On the other hand, several issues remain: a core issue is that reviewers remain unconvinced that NeuroFaith robustly measures the intended notion of faithfulness rather than a correlated effect. There is also a gap between the claims of the paper (this is related also to clarity and readability raised by one reviewer) and the conclusions reached experimentally and through ablations (see response by reviewer 2wXs, for example, as well as the largely unresolved issue about shortcut construction raised by reviewer LtSp).

**Reviewer Scores:**

Thanks to the rebuttal's additions I would expect limited increase in scores, perhaps by 2 points in total across all reviews (particularly for reviewers sWcB and 2wXs). There are some issues such as the one mentioned by reviewer 2wXs about metrics which would prevent a further increase in scores. And I also believe that a further discussion among reviewers to reach consensus might have made some reviewers more aware of the concerns raised by others (in particular several methodological concerns raised by reviewer LtSp).

So overall,  given the low starting scores and the fact that the reviewers remain somehow "unexcited" about this paper, I do not consider it meeting  the threshold for publication to ICLR.

---

### Decision · Program_Chairs · 2026-01-26

Reject